# Adding the temporal domain to PET radiomic features

**Wyanne A. Noortman**[1,2]*, **Dennis Vriens**[1], **Cornelis H. Slump**[3], **Johan Bussink**[4], **Tineke W. H. Meijer**[4], **Lioe-Fee de Geus-Oei**[1,2], **Floris H. P. van Velden**[1]

**1** Department of Radiology, Leiden University Medical Center, Leiden, The Netherlands, **2** Biomedical Photonic Imaging Group, University of Twente, Enschede, The Netherlands, **3** Robotics and Mechatronics, Technical Medical Centre, University of Twente, Enschede, The Netherlands, **4** Department of Radiation Oncology, Radboud University Medical Center, Nijmegen, The Netherlands

* w.a.noortman@lumc.nl

**Data Availability Statement:** All relevant data are within the manuscript and its Supporting Information files.

**Funding:** Dennis Vriens was supported in part by the Netherlands Organisation for Health Research

## Abstract

### Background

Radiomic features, extracted from positron emission tomography, aim to characterize tumour biology based on tracer intensity, tumour geometry and/or tracer uptake heterogeneity. Currently, radiomic features are derived from static images. However, temporal changes in tracer uptake might reveal new aspects of tumour biology. This study aims to explore additional information of these novel dynamic radiomic features compared to those derived from static or metabolic rate images.

### Methods

Thirty-five patients with non-small cell lung carcinoma underwent dynamic [18F]FDG PET/ CT scans. Spatial intensity, shape and texture radiomic features were derived from volumes of interest delineated on static PET and parametric metabolic rate PET. Dynamic grey level cooccurrence matrix (GLCM) and grey level run length matrix (GLRLM) features, assessing the temporal domain unidirectionally, were calculated on eight and sixteen time frames of equal length. Spearman's rank correlations of *parametric* and *dynamic* features with *static* features were calculated to identify features with potential additional information. Survival analysis was performed for the non-redundant temporal features and a selection of *static* features using Kaplan-Meier analysis.

### Results

Three out of 90 *parametric* features showed moderate correlations with corresponding *static* features ($\rho \geq 0.61$), all other features showed high correlations ($\rho > 0.7$). *Dynamic* features are robust independent of frame duration. Five out of 22 *dynamic* GLCM features showed a negligible to moderate correlation with any *static* feature, suggesting additional information. All sixteen *dynamic* GLRLM features showed high correlations with static features, implying redundancy. Log-rank analyses of Kaplan-Meier survival curves for all features dichotomised at the median were insignificant.

and Development (ZonMw) stipends for Qualified Doctor Training to become a Clinical Researcher (AGIKO) (project no. 92003552) for design and data collection of the original clinical study. The costs of the additional dynamic PET scans were covered by the Department of Radiology and Nuclear Medicine, Radboud University Medical Center, Nijmegen. There was no additional funding received for this study.

**Competing interests:** The authors have declared that no competing interests exist.

**Abbreviations:** 4D, four dimensional; CT, computed tomography; DCE, dynamic contrast enhanced; EANM, European Association of Nuclear Medicine; [18F]FDG, 2-[18F]fluoro-2-deoxy-D-glucose; FLAB, fuzzy locally adaptive Bayesian; GLCM, grey level cooccurrence matrix; GLDM, grey level dependence matrix; GLRLM, grey level run length matrix; GLSZM, grey level size zone matrix; IBSI, Image Biomarker Standardisation Initiative; IDMN, inverse difference moment normalized; IDN, inverse difference normalized; IMC, informational measure of correlation; $MR_{glc}$, parametric glucose metabolic rate; MRI, magnetic resonance imaging; OS, overall survival; NGTDM, neighbouring grey tone difference matrix; NSCLC, non-small cell lung carcinoma; PET, positron emission tomography; VOI, volume of interest.

## Conclusion

This study suggests that, compared to *static* features, some *dynamic* GLCM radiomic features show different information, whereas *parametric* features provide minimal additional information. Future studies should be conducted in larger populations to assess whether there is a clinical benefit of radiomics using the temporal domain over traditional radiomics.

## Introduction

In the field of radiomics, researchers aim to find stable and clinically relevant image-derived biomarkers, so-called radiomic features, that provide a non-invasive way of quantifying and monitoring tumour characteristics in clinical practice [1, 2]. For positron emission tomography (PET), radiomic features are investigated that quantify tracer intensity, tumour geometry and/or tracer uptake heterogeneity [3]. Tracer uptake heterogeneity, typically quantified using textural features, describing the spatial distribution of radiotracer uptake of the tumour, might provide information about cellular density, proliferation, angiogenesis, hypoxia, receptor expression, necrosis and fibrosis [4], and it is hypothesized that it, thereby, reflects specific regional variance in tumour characteristics including cancer genetics.

Traditional radiomic features describe heterogeneity along the *spatial* distribution of radiotracer uptake, but do not take into account tracer uptake heterogeneity over *time*, while this might contain additional information concerning tumour biology.

Research into these so-called temporal radiomics is limited. There are some studies that apply texture feature analysis on parametric images in magnetic resonance imaging (MRI) [5] and PET [6] using 3D images representing specific pharmacokinetic parameters. However, these studies did not assess time frames as the fourth dimension. To the best of our knowledge, there are no papers describing radiomics in PET using the temporal dimension.

Woods *et al.* have investigated the use of 4D texture analysis in dynamic contrast-enhanced MRI [7], but interchangeability of spatial and temporal dimensions was assumed. A different approach was found within proteomics, where Hu *et al.* studied the application of temporal texture features in time series fluorescence microscope images for the analysis of subcellular locations of proteins, since these location patterns indicate the possible function of a protein [8]. They investigated the original thirteen Haralick grey level cooccurrence matrix (GLCM) textural features in the temporal domain. Static GLCM features express combinations of grey levels of neighbouring pixels in the spatial domain [9]; the dynamic approach assesses adjacent voxels in time.

The current study explores whether the temporal domain reflects different aspects of tracer uptake and thereby tumour characteristics compared to static features. Inspired by the approach of Hu *et al.* [8], we used a different and novel approach, where texture features derived from the temporal domain using dynamic images are developed, i.e. dynamic GLCM features and dynamic grey level run length matrix (GLRLM) features. The aim of this study was to assess the potential additional information content of these *dynamic* texture features, comparing these to features derived from *parametric* images and to the more conventional spatial features derived from *static* images in NSCLC, thereby considering the temporal domain of tracer uptake.

## Materials and methods

### Patients and clinical follow-up

Dynamic 2-[18F]fluoro-2-deoxy-D-glucose ([18F]FDG) positron emission tomography scans combined with X-ray computed tomography (PET/CT) of a previously published prospective

cohort [10] were analysed. The study had been reviewed and approved by the Commission on Medical Research Involving Human Subjects Region Arnhem-Nijmegen, the Netherlands. All patients signed an informed consent form. The follow-up data of that previously published prospective cohort were updated. In short, the study included consecutive patients with newly diagnosed or suspected NSCLC, stage IB to stage IIIA, i.e. T2a-4N0-2M0 (TNM 7[th] edition), planned for primary resection in the Radboud University Medical Center between 2009 and 2014 [10]. All patients were routinely staged using contrast-enhanced CT of the chest and/or upper abdomen and [18F]FDG PET/CT with additional histologic staging of the mediastinum or other sites suspicious for cancer when necessary. Only tumours that were considered resectable, with a diameter larger than 30 mm were included to minimize the partial volume effect and be able to quantify heterogeneity [11]. Clinical characteristics of 35 included NSCLC lesions in 34 patients can be found in Table 1.

**Table 1. Clinical characteristics of 35 non-small cell lung carcinoma (NSCLC) lesions in 34 patients.**

| Characteristic | Value |
|---|---|
| **Age (years), median (range)** | 66 (44–80) |
| **Gender (M/F)** | 24/11 |
| **PET/CT scanner** | |
| **Biograph Duo** | 20 |
| **Biograph 40 mCT** | 15 |
| **$SUV_{max}$ (g.mL$^{-1}$), median (range)** | 13.63 (5.62–30.98) |
| **Histology** | |
| **Squamous cell carcinoma** | 20 |
| **Adenocarcinoma** | 12 |
| **Other** | 3 |
| **Differentiation** | |
| **Well differentiated** | 1 |
| **Moderately differentiated** | 13 |
| **Poorly differentiated** | 21 |
| **TNM stage (7th edition)** | |
| **Stage I** | 8 |
| **Stage II** | 20 |
| **Stage III** | 7 |
| **Surgical margins** | |
| **Free of tumour** | 32 |
| **Not free of tumour** | 2 |
| **Pathologic margins were inconclusive** | 1 |
| **Pleural invasion** | |
| **Yes** | 14 |
| **No** | 21 |
| **Adjuvant chemotherapy** | |
| **Yes** | 17 |
| **No** | 17 |
| **Unknown**/Lost to follow-up | 1 |
| **Adjuvant radiation therapy** | |
| **Yes** | 4 |
| **No** | 31 |
| **Median overall survival (months)** | 72 (95% CI: 49–95) |

## Patient preparation, data acquisition, image reconstruction and parametrisation

Within 7 days of surgery, patients underwent a dynamic [18F]FDG PET/CT scan with the primary tumour located centrally in the field of view using either the Biograph Duo (n = 21) or Biograph 40 mCT with TrueV z-axis gantry extension (n = 17) (Siemens Healthineers, Erlangen, Germany). Details on patient preparation, data acquisition and image reconstruction can be found in the original publication [10] and was in accordance with the European Association of Nuclear Medicine (EANM) guidelines for tumour imaging [12]. This resulted in acquisition of 70 time frames representing 60 min of tracer distribution, started at injection of [18F]FDG.

Static, parametric and dynamic volumes were used in this study and are illustrated in Fig 1. The final time frame (50–60 min p.i.) was used as static [18F]FDG PET image. The resulting voxel sizes were 2.56×2.56×3.38 and 1.59×1.59×2.03 mm$^3$ for the Biograph Duo PET/CT and Biograph 40 mCT PET/CT, respectively. Parametric glucose metabolic rate ($MR_{glc}$) images were computed based on image-derived tissue and blood time-activity concentration curves using Patlak method, with data acquired between 15 and 60 min normalized Patlak-time, using the Patlak slope ([18F]FDG influx constant, $K_i$), assuming a lumped constant of 1 and considering the plasma glucose concentration measured prior to [18F]FDG-injection [10].

Four dimensional (4D) dynamic volumes (x,y,z,t) with time frames of equal acquisition length were created combining the time frames between 10 and 50 min p.i. (16x 75 s; 8x 150 s). Equilibrium between perfusion and uptake of [18F]FDG is reached after 10–15 min of Patlak time, corresponding to approximately 10–15 min in real time [13]. Different lengths of time frames were created to assess dependence of radiomic features on frame duration, but the options were limited due to the unavailability of the raw data. The time frames were summed, resulting in sixteen 150 s-frames and eight 300 s-frames.

## Image analysis

Image acquisition, pre-processing and radiomic feature extraction were performed according to the Image Biomarker Standardisation Initiative (IBSI) guidelines. Details can be found in S1 File.

## Volumes of interest

Volumes of interest (VOI) of the tumour in the static and parametric images were drawn independently using a fuzzy locally adaptive Bayesian (FLAB) algorithm [14], excluding [18F]FDG-

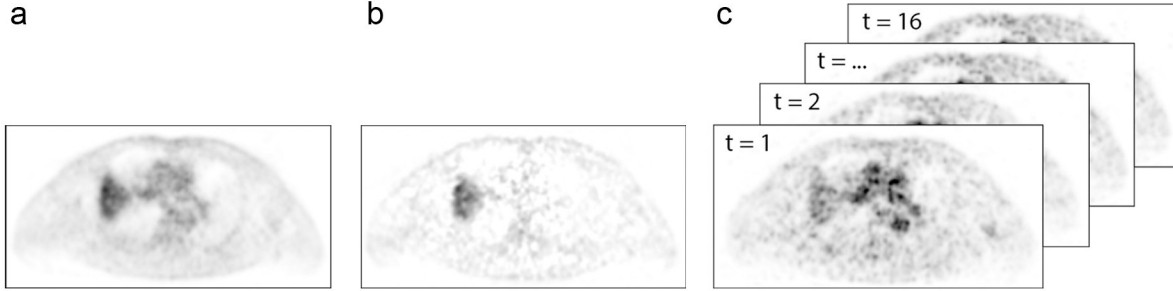

**Fig 1. Dynamic [18F]FDG PET acquisition of a patient with non-small cell lung carcinoma.** (A) The last time frame (50–60 min p.i.) used as static PET (i.e. voxel intensities represent standardised uptake values (SUV) [g.mL$^{-1}$]). (B) Parametric glucose metabolic rate images computed based on image-derived tissue- and blood time-activity concentration curves using Patlak method, with data acquired between 15 and 60 min normalized Patlak-time (i.e. voxel intensities represent [18F]FDG influx constants (min$^{-1}$)). (C) 4D dynamic volumes (x,y,z,t) consisting of 16 150 s-time frames acquired between 10 and 50 min p.i. (i.e. voxel intensities represent [18F]FDG activity-concentration at different time-points [Bq.mL$^{-1}$]).

avid non-tumour tissue by drawing an oversized container around the tumour and surrounding tissue by a radiation oncologist under supervision of an experienced nuclear medicine physician [10]. The VOIs of the static images were also used as VOIs for all the dynamic images.

## Interpolation and discretisation

Interpolation of the image and the VOI was performed to match the voxel sizes of both scanners and to create isotropic voxels, so that image matrices are rotationally invariant [15]. The slice thickness of the Biograph Duo was the largest spatial voxel dimension (3.38 mm). All images were interpolated as recommended by IBSI (trilinearly, grids aligned by centre) to this voxel dimension, i.e. $3.38 \times 3.38 \times 3.38$ mm$^3$ [15] using MATLAB 2017b (Mathworks, Natick, Massachusetts). Dynamic image frames were interpolated before they were combined to a 4D volume.

For the extraction of texture features, grey value discretisation was performed using a fixed bin width, since this leads to more robust features than a fixed number of bins [16]. For standardized uptake value-based images, a bin width of 0.5 g/mL has been described in literature [16]. To the best of our knowledge, optimal bin widths for parametric PET images are not known. Therefore, population-based bin widths were determined according to the Freedman-Diaconis rule [17]:

$$bin\ width = 2 \cdot IQR \cdot N^{-1/3} \tag{1}$$

with IQR the mean interquartile range and N the mean number of voxels in the VOIs of all included patients. Grey value discretisation of the dynamic images was performed using a fixed bin width for each of the individual time frames, calculated using Eq 1.

## Static and parametric features

Radiomic feature extraction of the static and parametric images was performed using PyRadiomics version 2.0 [18] in Python 3.6 (Python Software Foundation, Wilmington, Delaware). For every VOI, 90 features were calculated: intensity (18), shape (13), GLCM (22), GLRLM (16), grey level size zone matrix (GLSZM) (16) and neighbouring grey tone difference matrix (NGDTM) (5). Grey level dependence matrix (GLDM) features were not calculated, since these features are analogues to GLRLM and GLSZM features [15]. Image normalisation and distance weighting were not applied [15]. GLCM matrices were calculated assessing the VOIs in two directions per angle, thus taking into account rotational invariance [15]. GLCM and GLRLM matrices were calculated for thirteen angles (= $(3^3−1)/2$), corresponding to the corresponding direction vectors of the 26 directly neighbouring voxels within a neighbourhood volume at distance 1, and combined to one matrix [15].

## Dynamic features

Thirty-eight novel dynamic features were extracted from both the 150 s-frames and the 300 s-frames. Inspired by the approach of Hu *et al.* [8], novel dynamic texture features were developed that assess changes in voxel values of adjacent voxels in time, only regarding the temporal direction: $(x; y; z; t) = (0; 0; 0; 1)$. By including this temporal direction, the GLCM expresses if voxel values change from one time frame to the next and the GLRLM expresses the frequency of consecutive voxels with the same discretized grey level in the temporal dimension. When changes in tracer uptake over time are limited, the GLCM will only show non-zero values on and near the diagonal and the GLRM will show long runs. Due to causality in the temporal domain GLCMs were only calculated in one direction per angle instead of two [19]. Extraction

of 22 dynamic GLCM features and 16 dynamic GLRLM features on these temporal matrices was performed using PyRadiomics version 1.3 [18] in Python 3.6. Dynamic GLSZM and dynamic NGTDM features were not calculated, since these dynamic features would consider spatiotemporal directions instead of solely temporal directions, while spatial and temporal directions are not interchangeable [19]. No differences in implementation between PyRadiomics version 1.3 and 2.0, tested using example data, were found for the radiomic features investigated.

### Statistical analysis

Survival data are presented using Kaplan-Meier estimators. Overall survival is defined from date of the [$^{18}$F]FDG PET/CT to death of any cause, censoring all patient that were alive at the closeout date (July 30$^{th}$ 2018).

The dependence of the calculated features on the scanner was assessed using the GlobalTest package in R Statistical Software version 3.3.3 (R Foundation for Statistical Computing, Vienna, Austria) [20]. The association with the used scanner (Biograph Duo or Biograph 40 mCT) was assessed per feature set (static features (90), parametric features (90), dynamic features (38)) and for all features (218) using logistic regression with the features as covariates.

Robustness of dynamic features independent of the length of the time frames was assessed by calculating the Spearman's rank correlation coefficient ($\rho$) between the corresponding features from the 150 s- and the 300 s-frames in MATLAB. High correlations ($\rho>0.7$) indicate robustness of features independent of frame duration. Reproducibility would usually have been assessed using the intraclass correlation coefficient and Bland-Altman analysis, but due to size differences of the GLCM and GLRLM for the different frame durations, systematic *scale* errors would be expected. In contrast to systematic offset errors, these scale errors cannot be addressed using the conventional statistical methods.

Comparison of *parametric* and *dynamic* features with *static* features was performed using Spearman's rank correlation coefficient ($\rho$) in MATLAB. Parametric features with a high correlation (defined as $\rho>0.7$) with *corresponding* static features were considered redundant. Only correlations of corresponding features were calculated, since the mathematical definitions of static and parametric features are the same. Dynamic features with a high correlation with *any* static feature were considered redundant. These redundant features do not contain additional information compared to the original static spatial feature set.

To evaluate clinical relevance, the association of *static*, *parametric* and *dynamic* features with overall survival and histopathological characteristics was assessed in SPSS 23 (IBM Statistics, Chicago, IL). Parametric features and dynamic features that did not show high correlations with static features were assessed. Unsupervised feature selection of the static feature set was performed following the approach of Collarino *et al*. using redundancy filtering based on Pearson correlation coefficients ($r>0.75$) and principal component analysis (PCA) [21]. Survival curves were estimated using Kaplan-Meier analysis for the selected features dichotomized at their median and survival curves were compared using log-rank statistics. Differences in radiomic features between different binary histopathological characteristics were statistically compared using the Mann-Whitney U test or independent samples t-test, after testing for (log-)normality.

## Results

Extracted radiomic features and clinical data can be found in S1 Data.

### Bin widths

The bin width for static and parametric images was 0.55 g/mL and $1.8 \times 10^{-08}$ mol/mL/min, respectively. The bin widths for dynamic frames were 0.26, 0.29, 0.31, 0.34, 0.36, 0.36, 0.39,

0.40, 0.42, 0.44, 0.48, 0.49, 0.50, 0.51, 0.53, 0.54 g/mL for the 16 150 s-frames and 0.27, 0.31, 0.35, 0.39, 0.42, 0.48, 0.49, 0.53 g/mL for the 8 300 s-frames.

### Differences between scanners

No significant differences between the two scanners (Biograph Duo and Biograph 40 mCT) were found for the static features ($p = 0.069$), parametric features ($p = 0.145$), 150 s-frame dynamic features ($p = 0.077$) and all features together ($p = 0.088$).

### Parametric features

Eighty-seven out of 90 parametric features showed high correlations with corresponding static features, indicating redundancy. Only three out of 90 features did not show high correlations, but showed moderate correlations: short run low grey level emphasis (GLRLM) ($\rho = 0.693$, $P < 0.0001$), small area emphasis (GLSZM) ($\rho = 0.698$, $P < 0.0001$) and small area low grey level emphasis (GLSZM) ($\rho = 0.610$, $P = 0.0001$).

### Dynamic features

Thirty-two out of 38 dynamic features showed very high Spearman's rank correlations between 150 s- and 300 s-frames. Six GLCM features showed high correlations between frame lengths ($\rho \geq 0.757$): Correlation, informational measure of correlation 1 (IMC1), IMC2, inverse difference moment normalized (IDMN), inverse difference normalized (IDN) and inverse variance. Dynamic features were assumed to be robust to a change in frame durations, at least for those investigated in this study. Therefore, the following sections will only show the results for 150 s-frames.

Fig 2 shows the Spearman's rank correlation matrix of static and dynamic radiomic features derived from the 150 s-time frames. Five dynamic GLCM features show a negligible to moderate correlation ($\rho < 0.7$) with any static feature, indicating potential additional information to

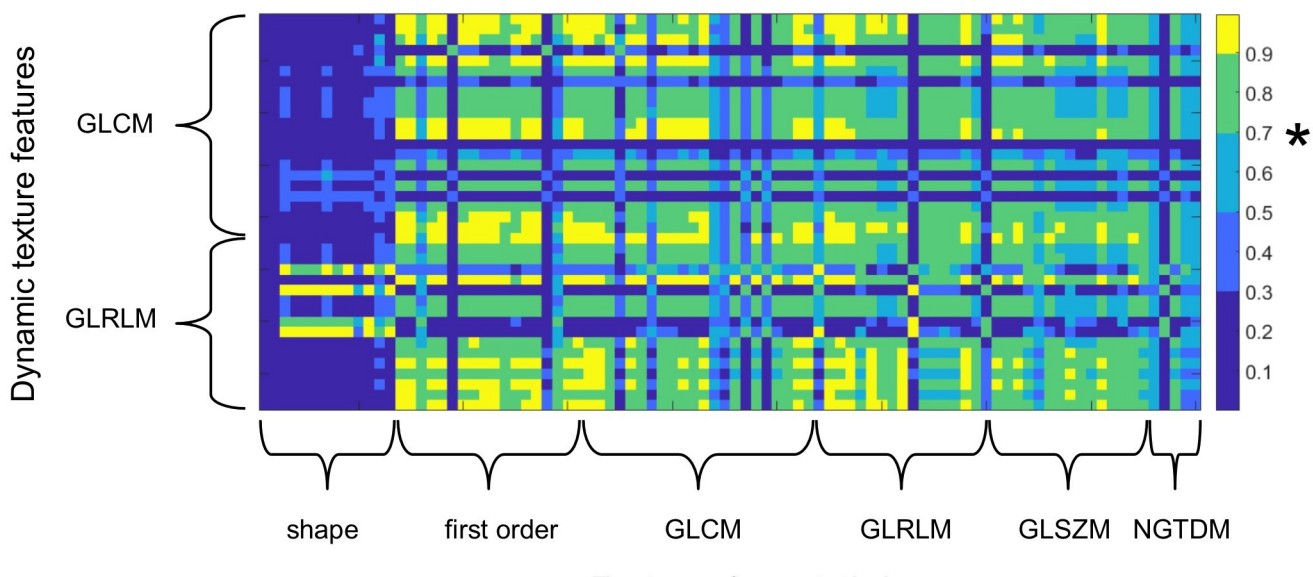

**Fig 2. Spearman correlation matrix of static radiomic features (x-axis) and dynamic texture features (y-axis).** Dynamic features with a maximum correlation with any static feature below 0.7 (marked in the colour bar by asterisk) contain additional information compared to static features. GLCM: grey level cooccurrence matrix, GLRLM: grey level run length matrix, GLSZM: grey level size zone matrix, NGTDM: neighbouring grey tone difference matrix.

existing features. The maximum correlation between any static feature and the dynamic GLCM features IMC1 and correlation were negligible to low ($\rho$ = 0.181 and = 0.483, respectively). Three features IDN, IDMN and IMC2 maximally showed a moderate correlation with any static features ($\rho$ = 0.533, $\rho$ = 0.542 and $\rho$ = 0.549, respectively). All other dynamic GLCM and dynamic GLRLM showed a high correlation ($\rho > 0.7$) with at least one static feature.

### Clinical relevance

PCA was inconclusive, since the number of subjects was too low compared to the number of features in order to obtain significant results. Therefore, the remaining five dynamic and three parametric features were compared to three traditional quantitative PET features: the maximum standardized uptake value (SUV$_{max}$), the metabolically active tumour volume (MTV) and the total lesion glycolysis (TLG). The estimated Kaplan-Meier survival curves for overall survival (OS) were not significantly different for these selected radiomic features dichotomized at their median. Kaplan-Meier survival curves can be found in S1 Fig and associations of radiomic features with histopathological characteristics can be found in S1 Table. The static SUV$_{max}$, the parametric short run low grey level emphasis (GLRLM) and the parametric small area low grey level emphasis (GLRLM) showed significant differences between adenocarcinomas and squamous cell carcinomas. Static MTV and TLG showed borderline significant differences in pleural invasion.

### Discussion

In this study, additional information and redundancy of novel radiomic features describing the temporal domain compared to traditional radiomic features derived from static images were investigated in a dataset of 35 lesions in 34 patients with stage IB to IIIA NSCLC who underwent a dynamic [18F]FDG PET/CT scan.

To the best of our knowledge, there are no papers describing radiomic features assessing tracer uptake heterogeneity over time in PET imaging. Woods *et al.* [7] investigated the addition of the temporal domain in radiomics. However, interchangeability of spatial and temporal dimensions was assumed, which might be questionable, since causality is a condition in the temporal dimension, while it is not in the spatial dimensions [19]. Therefore, inspired by the approach of Hu *et al.* in proteomics [8], where dynamic GLCM features were designed to assess combinations of grey levels of subsequent voxels in time, we developed novel dynamic GLCM and GLRLM features.

Five out of 22 *dynamic* GLCM features did not show redundancy in comparison to any *static* feature, suggesting potential additional information compared to static features. The *dynamic* GLRLM features showed high correlation with *static* features, demonstrating that these features likely do not include additional information concerning tracer uptake compared to the traditional radiomic feature set. Dynamic features were robust independent of frame length. Features from *parametric* images showed, except for three out of 90 features, high correlations with their *static* equivalents, demonstrating minimal additional information. These findings show that some dynamic GLCM features might potentially express additional information about tracer uptake, suggesting potential insights in tumour biology. Combined with the static feature set, they might provide extensive and in-depth knowledge in biological processes.

Unfortunately we could not show the clinical relevance using survival analysis in our small dataset. Parametric and dynamic as well as traditional quantitative PET features dichotomized at the median, did not show significant differences in Kaplan-Meier curves between the high and the low group. Intriguingly, even the quantitative feature SUV$_{max}$ did not show a difference, while high values of SUV$_{max}$ predicted a higher risk of death in patients with NSCLC [22], also in patients with the same stage as our cohort, treated with a surgical resection [23].

Two parametric features and the $SUV_{max}$ did show significant differences between adenocarcinomas and squamous cell carcinomas, but these features cannot be used for patient stratification, since the ranges of the features overlap. Since clinical added value could not be demonstrated, it is unknown whether the different information that the non-redundant features contained, has an added value interpreting tumour biology or consists of merely noise. Consequently, future dynamic PET studies with a large number of patients are warranted to investigate whether these dynamic radiomic features show any correlations with clinical outcome measures and are useful for predictive or prognostic purposes, especially, considering the development of more sophisticated machine learning and deep learning algorithms, even requiring larger numbers of patients [24].

In extension to Tixier *et al.* [6], correlations between radiomic features derived from static and parametric images were assessed side-by-side for this larger feature set. Correlations found by Tixier *et al.* cannot be compared one-to-one with our results, since different settings for radiomic feature extraction were used, e.g. a fixed number of bins versus a fixed bin width. Nevertheless, results of both studies show the same trend. All eight features assessed by Tixier *et al.* show high correlations between static and parametric radiomics. In our study, except for three features that were not assessed by Tixier *et al.*, all other parametric features showed a high correlation with their static equivalents. These three features showed moderate correlations ($\rho > 0.610$), only suggesting a minimal amount of additional information. It should be noted, though, that the study by Tixier *et al.* only included the first 20 patients of the cohort of the current study performed on a single PET-scanner (Biograph Duo).

Bin widths were calculated using the Freedman-Diaconis rule [17], since, to the best of our knowledge, bin widths for parametric images and dynamic frames cannot be found in literature. Another approach could be the calculation of the regression coefficient of the static $SUV_{mean}$ and mean glucose metabolic rate or $SUV_{mean}$ of a dynamic frame for all patients. This regression coefficient and a static bin width of 0.5 g/mL could be used to calculate the parametric and dynamic frame bin widths. In our population, the regression coefficient of the $SUV_{mean}$ and mean glucose metabolic rate was 0.0296 (strong linear correlation, r = 0.93). The static bin width of 0.55 g/mL, as calculated using the Freedman-Diaconis rule, would result in a bin width of 0.016 μmol/mL/min using the regression coefficient, which is very similar to the bin width of 0.018 μmol/mL/min calculated using the Freedman-Diaconis rule. The bin widths for the dynamic frames calculated using both methods were also similar.

Despite this paper lacking clinical relevance, in our opinion, the extraction of dynamic radiomic features might complement the static feature set and thereby advance the field by providing insight in different aspects of tumour biology. In addition, the field suffers from a publication bias with only 6% of radiomic papers describing negative results, as stretched by Buvat et al. [25]. Clinical validation and implementation in decision support of dynamic radiomic features might be difficult, as it requires relatively large datasets and dynamic PET acquisition is not standard-of-care due to high costs and invasive nature. This contradicts one of the main goals of radiomics of extracting biomarkers from standard-of-care medical images, lowering the need for additional or more invasive diagnostic methods. However, the recent introduction of total body PET scanners might facilitate acquisition of dynamic scans in clinical practice [26]. Also, extraction of texture features from dynamic images might be interesting in other modalities like dynamic contrast-enhanced (DCE) magnetic resonance imaging (MRI) and DCE-CT.

## Conclusion

In dynamic [$^{18}$F]FDG PET/CT scans in patients with non-small cell lung carcinoma, certain dynamic GLCM radiomic features show different information than traditional radiomic

features. These novel dynamic features are robust to an alternation in the frame duration. Features from parametric images only demonstrated minimal additional information. Future studies should assess whether there is a clinical benefit of radiomic features from dynamic images compared to traditional features derived from static images.

## Supporting information

**S1 File. Image biomarker standardisation initiative reporting guidelines.**
(DOCX)

**S1 Data. Data file radiomic features and clinical data.**
(XLSX)

**S1 Fig. Estimated Kaplan-Meier overall survival curves for selected features dichotomized at the median, compared using log-rank statistics.**
(DOCX)

**S1 Table. Association between histopathological characteristics and radiomic features calculated using the Mann-Whitney U test or independent samples t-test, after testing for (log-)normality.**
(DOCX)

## Acknowledgments

The authors want to thank Nicolle Peters for help with patient inclusion and Peter Kok and his team of PET technologists for support during acquisition of the original study.

## Author Contributions

**Conceptualization:** Wyanne A. Noortman, Dennis Vriens, Johan Bussink, Tineke W. H. Meijer, Lioe-Fee de Geus-Oei, Floris H. P. van Velden.

**Data curation:** Dennis Vriens, Johan Bussink, Tineke W. H. Meijer, Lioe-Fee de Geus-Oei.

**Formal analysis:** Wyanne A. Noortman, Dennis Vriens, Lioe-Fee de Geus-Oei, Floris H. P. van Velden.

**Investigation:** Wyanne A. Noortman, Dennis Vriens, Lioe-Fee de Geus-Oei, Floris H. P. van Velden.

**Methodology:** Wyanne A. Noortman, Dennis Vriens, Cornelis H. Slump, Lioe-Fee de Geus-Oei, Floris H. P. van Velden.

**Software:** Wyanne A. Noortman.

**Supervision:** Dennis Vriens, Cornelis H. Slump, Lioe-Fee de Geus-Oei, Floris H. P. van Velden.

**Writing – original draft:** Wyanne A. Noortman.

**Writing – review & editing:** Dennis Vriens, Cornelis H. Slump, Johan Bussink, Tineke W. H. Meijer, Lioe-Fee de Geus-Oei, Floris H. P. van Velden.

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
