## [Decision Letter · Decision Letter 0]

5 Jun 2020

PONE-D-20-05431

Adding the temporal domain to PET radiomic features

PLOS ONE

Dear Dr. Noortman,

Thank you for submitting your manuscript to PLOS ONE. After careful consideration, we feel that it has merit but does not fully meet PLOS ONE’s publication criteria as it currently stands. Therefore, we invite you to submit a revised version of the manuscript that addresses the points raised during the review process.

The study is interesting and might be important in this field. Please kindly address the issues from the reviewers.

We look forward to receiving your revised manuscript.

Kind regards,

Jason Chia-Hsun Hsieh, M.D. Ph.D

Academic Editor

PLOS ONE

Additional Editor Comments:

The study is interesting and might be important in this field.

2. In the ethics statement in the manuscript and in the online submission form, please provide additional information about the patient records used in your retrospective study, including:

a) whether all data were fully anonymized before you accessed them and

b) the date range (month and year) during which patients' medical records were accessed.

3. Please confirm in your methods section and ethics statement that the 'Commission on Medical Research Involving Human Subjects Region Arnhem-Nijmegen' consists of a committee of experts that reviewed and approved your study.

In addition, please clarify whether the present retrospective study was also granted ethical approval, in addition to the original study, and whether the previous prospective cohort were recruited by the same authors.

'Dennis Vriens was supported in part by the Netherlands Organisation for Health Research and Development (ZonMw) stipends for Qualified Doctor Training to become a Clinical Researcher (AGIKO) (project no. 92003552) for design and data collection of the original clinical study.'

a. Please provide an amended statement that declares *all* the funding or sources of support (whether external or internal to your organization) received during this study, as detailed online in our guide for authors at http://journals.plos.org/plosone/s/submit-now

Please also include the statement “There was no additional external funding received for this study.” in your updated Funding Statement.

5. Please include captions for your Supporting Information files at the end of your manuscript, and update any in-text citations to match accordingly. Please see our Supporting Information guidelines for more information: http://journals.plos.org/plosone/s/supporting-information

Reviewers' comments:

Reviewer's Responses to Questions

**Comments to the Author**

1. Is the manuscript technically sound, and do the data support the conclusions?

Reviewer #1: Yes

Reviewer #2: Yes

Reviewer #3: Yes

Reviewer #4: Yes

2. Has the statistical analysis been performed appropriately and rigorously? 

Reviewer #1: Yes

Reviewer #2: No

Reviewer #3: Yes

Reviewer #4: Yes

3. Have the authors made all data underlying the findings in their manuscript fully available?

Reviewer #1: Yes

Reviewer #2: Yes

Reviewer #3: No

Reviewer #4: No

4. Is the manuscript presented in an intelligible fashion and written in standard English?

Reviewer #1: Yes

Reviewer #2: Yes

Reviewer #3: Yes

Reviewer #4: Yes

5. Review Comments to the Author

Reviewer #1: The authors presented the possibility of adding PET radiomics features by additional information of temporal dynamic PET images. Currently, static information was used in the texture analysis for the radiomics analysis. However, kinetic information from the PET study may have significant data as a phenotyping information.

The authors describe the possibility of the use in PET radiomics using dynamic and kinetic data on PET.

They included only a 35-patients with NSCLC and dynamic FDG PET/CT, many kinds of additional parameters may have a additional information compared to the static PET images.

This is an preliminary report, but well-designed, prospective study with a meaningful suggestion for the use of total body PET with real dynamic PET study. Thus, this paper is very helpful for the development of imaging biomarker as well as PET radiomics phenotyping study.

Reviewer #2: This manuscript focuses on the development of a new method to evaluate the tumor heterogeneity based on dynamic 18F-FDG PET/CT images. Authors showed that new “dynamic” features can be extracted with a moderate correlation with “static” features.

Even if this new approach is promising, authors failed to demonstrate any added value. This may be because the cohort is too small or because the features do not carry information related to the patient survival. Worse, perhaps these new features do not reflect relevant biological information. Did the authors test the link between the “dynamic” features and the tumor histology or the differentiation for instance?

For the survival analysis, the authors do not seem to have taken into account the pleural invasion or the treatment (chemotherapy and/or radiation therapy), event though it is known that this strongly influences the prognosis.

This study may be too preliminary and requires further investigations to demonstrate a clinical benefit.

Reviewer #3: The paper addressed radiomics analysis on dynamic PET studies and include the temporal domain (either by using dynamic frames or using patlak Ki images). The paper is of interest as it generates some hypotheses and new ideas on the use of dynamic information for radiomics studies. As there is not much clinical benefit I would recommend to also describe the difference in RF values when analysed on Ki and SUV images (matching the number of bins used, see below)

Main comments:

VOIs – it seems that previously defined vois were reused or were these newly drawn?. Any differences in voi when defined on static and parametric images?. Did the authors redraw voi in the dynamic frames? If not, which one was used for dynamic analysis?

Bin width – I understand that for parametric images and for dynamic frames you cannot use the SUV=0.5 bin width, but you can estimate the slope between SUV and KI and then estimate how to convert SUV=0.5 bins into Ki bins. I recommend to do such an analysis to see if RF values become more comparable when you try to match the bin width (taking into account the different units of SUV and Ki).

Likewise, you can derive how many bins you got per static SUV image and use that number to process the dynamic frames or parametric images (for that patient).

Apart from assessing spearman correlations, it would be nice to demonstrate the difference in RF values, eg by using a distance metric or else, but it will likely require the above suggested bin width adaptions as well.

You have mainly stage 2 subjects. There is no difference in KM plots. Maybe add some case control evaluations by looking at the feature values for stage 1 versus stage 3 and see if there are any features that are significantly different between these 2 more extreme cases…I realize it is only about 7 subjects per group (in this case), but it would give an hypothesis if some features might have prognostic value?

Likewise, you can take the 25% short survivors and 25% long survivors and see if there is any difference between these groups.

Minor comments:

Abstract,results: 3 out of 90 show moderate correlation, so the others were highly correlated. please state so.

Two scanners are used. Was there any cross-calibration between the systems. They do not find significant differences (yet a trend, so there must be some difference there?).

Patlak images – which software was used?

2 software packages were used for RF. Any chance of different implementation issues? Did the authors compare results from the 2 packages.

Reviewer #4: In the manuscript

„Adding the temporal domain to PET radiomic features”,

the authors analyze the effect of introducing the temporal domain in the assessment of radiomics features in PET data. This is a new and innovative idea in radiomics analysis of PET data and of high interest for the community. The manuscript is very well written and good to understand. I just have some minor issues that should be changes:

- How were the blood time-activity concentration curves estimated, by blood-sampling or image-based; if so, was the left ventricle used?

- The resolution of figure 4 is way to low, it is hardly possible to see anything. Pleas also include the risk tables below the Kaplan-Meier curves.

6. PLOS authors have the option to publish the peer review history of their article (what does this mean?). If published, this will include your full peer review and any attached files.

Reviewer #1: No

Reviewer #2: No

Reviewer #3: No

Reviewer #4: No

---

## [Author Response · Author response to Decision Letter 0]

23 Jul 2020

Article ID: PONE-D-20-05431

Title: Adding the temporal domain to PET radiomic features

We would like to thank the reviewers for their helpful comments which have improved the quality of the paper. The specific comments of the reviewers are in blue/italic below together with their responses. Unfortunately, we did not manage to get the text in blue/italic in this editor, but this version can be found in the attached files. We have changed the manuscript accordingly (changes are marked in red).

Response to Journal requirements:

We apologize for missing these files when submitting the manuscript. The manuscript is now updated according to the style requirements.

2. In the ethics statement in the manuscript and in the online submission form, please provide additional information about the patient records used in your retrospective study, including:

a) whether all data were fully anonymized before you accessed them and

b) the date range (month and year) during which patients' medical records were accessed.

After informed consent, relevant patient data were included in the case report forms by the treating physicians. Follow-up data were updated in July 2018. Clinical data were pseudonymized before radiomic analysis. Imaging data (DICOM files) were left unchanged, as advanced quantitative analysis was part of the original study for which the patients provided written informed consent and as DICOM anonymization might remove DICOM tags that are crucial for absolute [18F]FDG quantification. After data extraction all further analyses were performed pseudonymized. Imaging data were accessed between January and October 2018. Pseudonymized clinical data and radiomic features were assessed between January 2018 and February 2020.

3. Please confirm in your methods section and ethics statement that the 'Commission on Medical Research Involving Human Subjects Region Arnhem-Nijmegen' consists of a committee of experts that reviewed and approved your study.

In addition, please clarify whether the present retrospective study was also granted ethical approval, in addition to the original study, and whether the previous prospective cohort were recruited by the same authors.

In the present study, we performed an additional analysis of this previously published cohort. For the current study ethical approval was granted within the approval of the original prospective study. The application included quantitative image analysis, as implemented in this study. The prospective cohort was recruited by the same authors (DV, JB, TM and LF).

'Dennis Vriens was supported in part by the Netherlands Organisation for Health Research and Development (ZonMw) stipends for Qualified Doctor Training to become a Clinical Researcher (AGIKO) (project no. 92003552) for design and data collection of the original clinical study.'

 a. Please provide an amended statement that declares *all* the funding or sources of support (whether external or internal to your organization) received during this study, as detailed online in our guide for authors at http://journals.plos.org/plosone/s/submit-now

Please also include the statement “There was no additional external funding received for this study.” in your updated Funding Statement.

 Thank you, the funding statement was changed.

5. Please include captions for your Supporting Information files at the end of your manuscript, and update any in-text citations to match accordingly. Please see our Supporting Information guidelines for more information: http://journals.plos.org/plosone/s/supporting-information

Thank you for pointing this out, the supporting information files were added as captions at the end of the manuscript, in line with the Supporting Information guidelines.

 

Response to comments to the author:

Comments to the Author

1. Is the manuscript technically sound, and do the data support the conclusions?

Reviewer #1: Yes

Reviewer #2: Yes

Reviewer #3: Yes

Reviewer #4: Yes

2. Has the statistical analysis been performed appropriately and rigorously? 

Reviewer #1: Yes

Reviewer #2: No

Reviewer #3: Yes

Reviewer #4: Yes

3. Have the authors made all data underlying the findings in their manuscript fully available?

Reviewer #1: Yes

Reviewer #2: Yes

Reviewer #3: No

Reviewer #4: No

4. Is the manuscript presented in an intelligible fashion and written in standard English?

Reviewer #1: Yes

Reviewer #2: Yes

Reviewer #3: Yes

Reviewer #4: Yes

 

5. Review Comments to the Author

Reviewer #1: The authors presented the possibility of adding PET radiomics features by additional information of temporal dynamic PET images. Currently, static information was used in the texture analysis for the radiomics analysis. However, kinetic information from the PET study may have significant data as a phenotyping information.

The authors describe the possibility of the use in PET radiomics using dynamic and kinetic data on PET.

They included only a 35-patients with NSCLC and dynamic FDG PET/CT, many kinds of additional parameters may have a additional information compared to the static PET images.

This is an preliminary report, but well-designed, prospective study with a meaningful suggestion for the use of total body PET with real dynamic PET study. Thus, this paper is very helpful for the development of imaging biomarker as well as PET radiomics phenotyping study.

We would like to thank the reviewer for the compliments and endorsement of the potential benefit of these novel dynamic radiomics features in future total body PET studies.

 

Reviewer #2: This manuscript focuses on the development of a new method to evaluate the tumor heterogeneity based on dynamic 18F-FDG PET/CT images. Authors showed that new “dynamic” features can be extracted with a moderate correlation with “static” features.

Even if this new approach is promising, authors failed to demonstrate any added value. This may be because the cohort is too small or because the features do not carry information related to the patient survival. Worse, perhaps these new features do not reflect relevant biological information. Did the authors test the link between the “dynamic” features and the tumor histology or the differentiation for instance?

We would like to thank the reviewer for this suggestion. Unfortunately, homogeneous and large cohorts of patients who underwent dynamic PET scans are scarce. We investigated the association of the parametric, dynamic and traditional quantitative PET features with clinical parameters. We added supporting file 3 to the manuscript, where these findings are presented. Table 1 of S3 presents the association with clinical parameters. Two parametric features and the SUVmax showed significant differences in mean between adenocarcinoma and squamous cell carcinoma, but the ranges of the values are overlapping, indicating that these parameters cannot be used to discriminate between both histopathological subtypes. These findings were added to the manuscript (lines: 236-238, 287-290, 331-333).

For the survival analysis, the authors do not seem to have taken into account the pleural invasion or the treatment (chemotherapy and/or radiation therapy), event though it is known that this strongly influences the prognosis.

The reviewer is correct. We did not take into account these factors, because we only performed Kaplan-Meier analysis. In table 1 we added the univariate and multivariate Cox regression analysis of the clinical characteristics. In our population, only one variable, the age of diagnosis, is significantly associated with survival. 

Table 1: Univariate and multivariate Cox regression analysis of clinical characteristics, traditional quantitative PET features and selected radiomic features for overall survival (OS), disease-free survival (DFS) and disease-specific survival (DSS). Characteristics and features with a p-value < 0.20 in univariate analysis, were selected for forward and backward multivariate analysis based on the likelihood ratio. Pleural invasion and adjuvant chemotherapy were removed from the model in both forward and backward multivariate Cox regression.

 Hazard ratio (95% confidence interval) p-value

Univariate Cox regression analysis

Gender (male/female) 1.937 (0.545 - 6.889) 0.298

Age of diagnosis (years) 1.086 (1.014 - 1.164) 0.015

Stadium 

 IB 1.000 0.697

 IIA 1.514 (0.252 - 9.100) 

 IIB 1.081 (0.198 - 5.922) 

 IIIA 2.233 (0.449 - 11.112) 

Pleural invasion 0.486 (1.175 - 1.349) 0.157

Negative resection margin 0.594 (0.074 - 4.756) 0.620

Adjuvant chemotherapy 2.390 (0.796 - 7.181) 0.109

Post-operative radiotherapy 0.849 (0.189 - 3.809) 0.830

Traditional quantitative PET features

SUVmax (g/mL) 1.029 (0.957 - 1.106) 0.432

MTV (mL) 1.000 (1.000 - 1.000) 0.606

TLG (g) 1.000 (1.000 - 1.000) 0.611

Parametric features 

GLRLM SRLGLE 5.645 (0.000 - 94206.982) 0.727

GLSZM SAE 17.494 (0.099 - 3083.885) 0.279

GLSZM SALGLE 1.615 (0.016 - 160.357) 0.838

Dynamic GLCM 

Correlation 0.253 (0.000 - 1287.290) 0.253

IMC1 18.304 (0.002 - 149105.596) 0.526

IMC2 1.925 (0.000 - 182838.748) 0.911

IDMN 0.000 (0.000 - 4.23E+73) 0.714

IDN 0.000 (0.000 - 2.339E+17) 0.663

Multivariate Cox regression analysis

Iterative forward selection 

Age of diagnosis (years) 1.083 (1.009 - 1.163) 0.027

Pleural invasion 

Adjuvant chemotherapy 

Iterative backward selection

Age of diagnosis (years) 1.083 (1.009 - 1.163) 0.027

Pleural invasion 

Adjuvant chemotherapy 

This study may be too preliminary and requires further investigations to demonstrate a clinical benefit.

We would like to thank the reviewer for the helpful suggestions and hope that by adding these additional analyses to the manuscript we have improved its quality. We agree that the study is too preliminary to show clinical benefit and that this study only shows that some dynamic GLCM radiomic features show different information. We further agree that future studies with a larger cohort should be conducted to show the additional clinical benefit of these dynamic features. This is acknowledged in the discussion.

 

Reviewer #3: The paper addressed radiomics analysis on dynamic PET studies and include the temporal domain (either by using dynamic frames or using patlak Ki images). The paper is of interest as it generates some hypotheses and new ideas on the use of dynamic information for radiomics studies. As there is not much clinical benefit I would recommend to also describe the difference in RF values when analysed on Ki and SUV images (matching the number of bins used, see below)

We would like to thank the reviewer for the helpful recommendations. We hope to have sufficiently addressed all of the reviewers concerns below.

Main comments:

VOIs – it seems that previously defined vois were reused or were these newly drawn?. Any differences in voi when defined on static and parametric images?. Did the authors redraw voi in the dynamic frames? If not, which one was used for dynamic analysis?

The reviewer is correct, we reused the VOIs that were previously defined in the original study. These were defined on the static images and parametric images separately. For the current analysis, we copied them unchanged to the static and parametric images. To avoid that the delineation would change between different frames and thereby impact feature extraction, we reused the static VOIs as VOIs for the dynamic frames. So these VOIs were also not redrawn. This can be found in lines 152-156 of the manuscript.

Bin width – I understand that for parametric images and for dynamic frames you cannot use the SUV=0.5 bin width, but you can estimate the slope between SUV and KI and then estimate how to convert SUV=0.5 bins into Ki bins. I recommend to do such an analysis to see if RF values become more comparable when you try to match the bin width (taking into account the different units of SUV and Ki).

Likewise, you can derive how many bins you got per static SUV image and use that number to process the dynamic frames or parametric images (for that patient).

Thank you for the interesting suggestion, we did not consider this. The graph in figure 1 (left) shows how the SUVmean translates to the mean glucose metabolic rate of the parametric images for all patients in our study. We used the slope of this graph to calculate the parametric bin width, which would be 0.0296*0.5 = 0.016 µmol/mL/min. We used the same approach for the dynamic frames, as an example, figure 1 (right) shows how the static SUVmean translates to the SUVmean in frame 62, which would result in a bin width of 0.7038*0.5 = 0.35 g/mL. We did this for all dynamic frames. It turns out that these bin widths are very similar to the bin widths calculated with the Freedman-Diaconis rule, especially, when we calculate the bin widths using the slope and a static bin width of 0.55 g/mL, as used in our study. Table 2 shows the calculated slopes and bin widths and the bin widths calculated with the Freedman-Diaconis rule, for all used images. Since these values are quite similar, we did not change the bin widths, but we did mention this in our manuscript (lines: 354-363).

Figure 1. Right: Calculation of the slope of the static SUVmean and the mean glucose metabolic rate for all subjects. Left: Calculation of the slope of the static SUVmean and the SUVmean of dynamic frame 62. The slopes can be used to translate the bin width of the static images to parametric and dynamic bin widths, respectively.

Table 2: Calculated slopes and slope bin widths and bin widths calculated using the Freedman-Diaconis rule.

Image Static MRglu F46 F48 F50 F52 F54 F56 F58 F60

Slope 0.0296 0.4317 0.4666 0.5017 0.5442 0.5723 0.606 0.6434 0.6755

Bin width slope 0.55 0.016 0.24 0.26 0.28 0.30 0.32 0.33 0.35 0.37

Bin width Freedman-Diaconis 0.55 0.018 0.26 0.29 0.31 0.34 0.36 0.36 0.37 0.4

Image F62 F63 F64 F65 F66 F67 F68 F69

Slope 0.7038 0.7368 0.7542 0.7794 0.8025 0.8189 0.856 0.8814

Bin width slope 0.39 0.40 0.42 0.43 0.44 0.45 0.47 0.49

Bin width Freedman-Diaconis 0.42 0.44 0.48 0.49 0.5 0.51 0.53 0.54

Apart from assessing spearman correlations, it would be nice to demonstrate the difference in RF values, eg by using a distance metric or else, but it will likely require the above suggested bin width adaptions as well.

In the past we have investigated comparing features using distance metrics, but it is difficult to interpret these results for the different radiomic features, since mathematical definitions vary largely from feature to feature. 

You have mainly stage 2 subjects. There is no difference in KM plots. Maybe add some case control evaluations by looking at the feature values for stage 1 versus stage 3 and see if there are any features that are significantly different between these 2 more extreme cases…I realize it is only about 7 subjects per group (in this case), but it would give an hypothesis if some features might have prognostic value?

Likewise, you can take the 25% short survivors and 25% long survivors and see if there is any difference between these groups.

We would like to thank the reviewer for this suggestion. We added supporting file 3 to the manuscript. Table 1 of S3 presents the association with clinical parameters. Unfortunately, no significant differences were found.

Minor comments:

Abstract,results: 3 out of 90 show moderate correlation, so the others were highly correlated. please state so.

We added this for clarity (line 43). 

Two scanners are used. Was there any cross-calibration between the systems. They do not find significant differences (yet a trend, so there must be some difference there?).

The scanners were not used at the same time, so cross-calibration between the scanners has not been performed. However, both scanners were EARL accredited and were cross-calibrated with the dose calibrator. The same dose calibrator was used throughout the whole study. 

Patlak images – which software was used?

The Patlak analysis was performed in Inveon Research Workplace (Siemens Healthineers, Erlangen, Germany). This was added to Appendix S1, which contains more information on image acquisition and reconstructions.

2 software packages were used for RF. Any chance of different implementation issues? Did the authors compare results from the 2 packages.

The main difference between PyRadiomics 1.3 and 2.0 is the calculation of the matrices from which the features are extracted. In PyRadiomics 1.3, these matrices are calculated in Python, while in PyRadiomics 2.0, the matrices are calculated in C. We adjusted PyRadiomics 1.3 for the extraction of the dynamic features. For comparison, table 3 presents radiomic features calculated for the PyRadiomics example data, for version 1.3 and 2.0. No differences were found between both implementations. This was added in line 200 of the manuscript.

Table 3. Feature values for PyRadiomics version 1.3 and 2.0 and differences between implementations.

Feature Version 1.3 Version 2.0 Difference

Image lung1_image.nrrd lung1_image.nrrd 

Mask lung1_label.nrrd lung1_label.nrrd 

general_info_BoundingBox (206, 347, 32, 24, 26, 3) (206, 347, 32, 24, 26, 3) 

general_info_EnabledImageTypes {'Original': {}} {'Original': {}} 

general_info_GeneralSettings {'minimumROIDimensions': 1, 'minimumROISize': None, 'normalize': False, 'normalizeScale': 1, 'removeOutliers': None, 'resampledPixelSpacing': None, 'interpolator': 'sitkBSpline', 'preCrop': False, 'padDistance': 5, 'distances': [1], 'force2D': False, 'force2Ddimension': 0, 'resegmentRange': None, 'label': 1, 'additionalInfo': True, 'voxelBased': False} {'minimumROIDimensions': 1, 'minimumROISize': None, 'normalize': False, 'normalizeScale': 1, 'removeOutliers': None, 'resampledPixelSpacing': None, 'interpolator': 'sitkBSpline', 'preCrop': False, 'padDistance': 5, 'distances': 

general_info_ImageHash 34dca4200809a5e76c702d6b9503d958093057a3 34dca4200809a5e76c702d6b9503d958093057a3 

general_info_ImageSpacing (0.5703125, 0.5703125, 5.0) (0.5703125, 0.5703125, 5.0) 

general_info_MaskHash 054d887740012177bd1f9031ddac2b67170af0f3 054d887740012177bd1f9031ddac2b67170af0f3 

general_info_NumpyVersion 1.19.0 1.19.0 

general_info_PyWaveletVersion 1.1.1 1.1.1 

general_info_SimpleITKVersion 1.2.4 1.2.4 

general_info_Version 1.3.0 2.0.0 

general_info_VolumeNum 1 1 

general_info_VoxelNum 837 837 

original_shape_Elongation 0.718791031 0.718791031 0

original_shape_Flatness 0.514335768 0.514335768 0

original_shape_LeastAxis 8.936318224 8.936318224 0

original_shape_MajorAxis 17.37448332 17.37448332 0

original_shape_Maximum2DDiameterColumn 16.04444054 16.04444054 0

original_shape_Maximum2DDiameterRow 13.53756348 13.53756348 0

original_shape_Maximum2DDiameterSlice 15.97893091 15.97893091 0

original_shape_Maximum3DDiameter 18.18259471 18.18259471 0

original_shape_MinorAxis 12.48862278 12.48862278 0

original_shape_Sphericity 0.75931875 0.75931875 0

original_shape_SurfaceArea 782.241458 782.241458 0

original_shape_SurfaceVolumeRatio 0.574671403 0.574671403 0

original_shape_Volume 1361.197815 1361.197815 0

original_firstorder_10Percentile -245.4 -245.4 0

original_firstorder_90Percentile 71 71 0

original_firstorder_Energy 16291991 16291991 0

original_firstorder_Entropy 4.020834927 4.020834927 0

original_firstorder_InterquartileRange 198 198 0

original_firstorder_Kurtosis 2.695927096 2.695927096 0

original_firstorder_Maximum 106 106 0

original_firstorder_MeanAbsoluteDeviation 105.0944475 105.0944475 0

original_firstorder_Mean -63.9080048 -63.9080048 0

original_firstorder_Median -31 -31 0

original_firstorder_Minimum -506 -506 0

original_firstorder_Range 612 612 0

original_firstorder_RobustMeanAbsoluteDeviation 81.58090535 81.58090535 0

original_firstorder_RootMeanSquared 139.5161078 139.5161078 0

original_firstorder_Skewness -0.73366595 -0.73366595 0

original_firstorder_TotalEnergy 26495367.44 26495367.44 0

original_firstorder_Uniformity 0.074426645 0.074426645 0

original_firstorder_Variance 15380.51125 15380.51125 0

original_glcm_Autocorrelation 411.4164748 411.4164748 0

original_glcm_ClusterProminence 9732.694396 9732.694396 0

original_glcm_ClusterShade -345.713367 -345.713367 0

original_glcm_ClusterTendency 58.74756668 58.74756668 0

original_glcm_Contrast 20.7134493 20.7134493 0

original_glcm_Correlation 0.470613617 0.470613617 0

original_glcm_DifferenceAverage 3.216603092 3.216603092 0

original_glcm_DifferenceEntropy 3.187524502 3.187524502 0

original_glcm_DifferenceVariance 9.381995813 9.381995813 0

original_glcm_Id 0.417361964 0.417361964 0

original_glcm_Idm 0.344350177 0.344350177 0

original_glcm_Idmn 0.972582394 0.972582394 0

original_glcm_Idn 0.899630707 0.899630707 0

original_glcm_Imc1 -0.17331187 -0.17331187 0

original_glcm_Imc2 0.818766382 0.818766382 0

original_glcm_InverseVariance 0.278697167 0.278697167 0

original_glcm_JointAverage 20.04512484 20.04512484 0

original_glcm_JointEnergy 0.017918271 0.017918271 0

original_glcm_JointEntropy 6.932828996 6.932828996 0

original_glcm_MaximumProbability 0.089125606 0.089125606 0

original_glcm_SumAverage 40.09024968 40.09024968 0

original_glcm_SumEntropy 4.635501946 4.635501946 0

original_glcm_SumSquares 19.865254 19.865254 0

original_gldm_DependenceEntropy 6.550399892 6.550399892 0

original_gldm_DependenceNonUniformity 120.9761051 120.9761051 0

original_gldm_DependenceNonUniformityNormalized 0.144535371 0.144535371 0

original_gldm_DependenceVariance 18.00291477 18.00291477 0

original_gldm_GrayLevelNonUniformity 62.29510155 62.29510155 0

original_gldm_GrayLevelVariance 24.73367791 24.73367791 0

original_gldm_HighGrayLevelEmphasis 383.9199522 383.9199522 0

original_gldm_LargeDependenceEmphasis 37.44922342 37.44922342 0

original_gldm_LargeDependenceHighGrayLevelEmphasis 20425.03584 20425.03584 0

original_gldm_LargeDependenceLowGrayLevelEmphasis 0.075562484 0.075562484 0

original_gldm_LowGrayLevelEmphasis 0.005605862 0.005605862 0

original_gldm_SmallDependenceEmphasis 0.318186003 0.318186003 0

original_gldm_SmallDependenceHighGrayLevelEmphasis 84.05116859 84.05116859 0

original_gldm_SmallDependenceLowGrayLevelEmphasis 0.003630175 0.003630175 0

original_glrlm_GrayLevelNonUniformity 48.265238 48.265238 0

original_glrlm_GrayLevelNonUniformityNormalized 0.066018368 0.066018368 0

original_glrlm_GrayLevelVariance 24.66124095 24.66124095 0

original_glrlm_HighGrayLevelRunEmphasis 362.3993952 362.3993952 0

original_glrlm_LongRunEmphasis 1.756790194 1.756790194 0

original_glrlm_LongRunHighGrayLevelEmphasis 758.781125 758.781125 0

original_glrlm_LongRunLowGrayLevelEmphasis 0.007773425 0.007773425 0

original_glrlm_LowGrayLevelRunEmphasis 0.006164929 0.006164929 0

original_glrlm_RunEntropy 4.555631762 4.555631762 0

original_glrlm_RunLengthNonUniformity 602.3643647 602.3643647 0

original_glrlm_RunLengthNonUniformityNormalized 0.819694212 0.819694212 0

original_glrlm_RunPercentage 0.868853966 0.868853966 0

original_glrlm_RunVariance 0.382061813 0.382061813 0

original_glrlm_ShortRunEmphasis 0.920028572 0.920028572 0

original_glrlm_ShortRunHighGrayLevelEmphasis 322.2128305 322.2128305 0

original_glrlm_ShortRunLowGrayLevelEmphasis 0.005979334 0.005979334 0

original_glszm_GrayLevelNonUniformity 18.29530201 18.29530201 0

original_glszm_GrayLevelNonUniformityNormalized 0.061393631 0.061393631 0

original_glszm_GrayLevelVariance 21.55781271 21.55781271 0

original_glszm_HighGrayLevelZoneEmphasis 262.7449664 262.7449664 0

original_glszm_LargeAreaEmphasis 93.21812081 93.21812081 0

original_glszm_LargeAreaHighGrayLevelEmphasis 51136.9698 51136.9698 0

original_glszm_LargeAreaLowGrayLevelEmphasis 0.184875307 0.184875307 0

original_glszm_LowGrayLevelZoneEmphasis 0.010725736 0.010725736 0

original_glszm_SizeZoneNonUniformity 138.7248322 138.7248322 0

original_glszm_SizeZoneNonUniformityNormalized 0.465519571 0.465519571 0

original_glszm_SmallAreaEmphasis 0.709478222 0.709478222 0

original_glszm_SmallAreaHighGrayLevelEmphasis 170.3079618 170.3079618 0

original_glszm_SmallAreaLowGrayLevelEmphasis 0.009509532 0.009509532 0

original_glszm_ZoneEntropy 5.514483642 5.514483642 0

original_glszm_ZonePercentage 0.356033453 0.356033453 0

original_glszm_ZoneVariance 85.32918562 85.32918562 0

original_ngtdm_Busyness 0.19930534 0.19930534 0

original_ngtdm_Coarseness 0.008985148 0.008985148 0

original_ngtdm_Complexity 617.8974921 617.8974921 0

original_ngtdm_Contrast 0.183769662 0.183769662 0

original_ngtdm_Strength 2.786428904 2.786428904 0

 

Reviewer #4: In the manuscript

„Adding the temporal domain to PET radiomic features”,

the authors analyze the effect of introducing the temporal domain in the assessment of radiomics features in PET data. This is a new and innovative idea in radiomics analysis of PET data and of high interest for the community. The manuscript is very well written and good to understand. I just have some minor issues that should be changes:

We would like to thank the reviewer for this positive feedback.

- How were the blood time-activity concentration curves estimated, by blood-sampling or image-based; if so, was the left ventricle used?

We would like to thank the reviewer for this suggestion, we indeed forgot to mention this in the manuscript. The image-derived input function was based on a 10 mL VOI of the descending aorta on which endothelial wall and calcifications were excluded to identify only blood, drawn on the images obtained during the first 60 seconds. We added this to S1, which contains more information on image acquisition and reconstruction.

- The resolution of figure 4 is way to low, it is hardly possible to see anything. Pleas also include the risk tables below the Kaplan-Meier curves.

We thank the reviewer for pointing this out, we have resolved this issue by using a larger font size and larger images. Also, the survival curves have been moved to an additional supporting information file (S3), combined with the association of radiomic features with other clinical parameters.

6. PLOS authors have the option to publish the peer review history of their article (what does this mean?). If published, this will include your full peer review and any attached files.

Do you want your identity to be public for this peer review? For information about this choice, including consent withdrawal, please see our Privacy Policy.

Reviewer #1: No

Reviewer #2: No

Reviewer #3: No

Reviewer #4: No

---

## [Decision Letter · Decision Letter 1]

7 Sep 2020

Adding the temporal domain to PET radiomic features

PONE-D-20-05431R1

Dear Dr. Noortman,

We’re pleased to inform you that your manuscript has been judged scientifically suitable for publication and will be formally accepted for publication once it meets all outstanding technical requirements.

Kind regards,

Jason Chia-Hsun Hsieh, M.D. Ph.D

Academic Editor

PLOS ONE

Additional Editor Comments (optional):

Most of the questions were answered adequately.

Reviewers' comments:

Reviewer's Responses to Questions

**Comments to the Author**

1. If the authors have adequately addressed your comments raised in a previous round of review and you feel that this manuscript is now acceptable for publication, you may indicate that here to bypass the “Comments to the Author” section, enter your conflict of interest statement in the “Confidential to Editor” section, and submit your "Accept" recommendation.

Reviewer #3: All comments have been addressed

2. Is the manuscript technically sound, and do the data support the conclusions?

Reviewer #3: Yes

3. Has the statistical analysis been performed appropriately and rigorously? 

Reviewer #3: N/A

4. Have the authors made all data underlying the findings in their manuscript fully available?

Reviewer #3: Yes

5. Is the manuscript presented in an intelligible fashion and written in standard English?

Reviewer #3: Yes

6. Review Comments to the Author

Reviewer #3: The authors have addressed all concerns/comments. I have no further comments.

One minor issue: will we ever have enough dynamic FDG studies to assess the clinical value for dynamic radiomics? Maybe a comment in line with recommendation of Rich Carson to do a dynamic whole body scan for every first patient of the day could be made. The first hour pi (uptake time) is otherwise not used anyway.

7. PLOS authors have the option to publish the peer review history of their article (what does this mean?). If published, this will include your full peer review and any attached files.

Reviewer #3: No

---

## [Editor Report · Acceptance letter]

15 Sep 2020

PONE-D-20-05431R1 

Adding the temporal domain to PET radiomic features 

Dear Dr. Noortman:

I'm pleased to inform you that your manuscript has been deemed suitable for publication in PLOS ONE. Congratulations! Your manuscript is now with our production department. 

Kind regards, 

on behalf of

Dr. Jason Chia-Hsun Hsieh 

Academic Editor

PLOS ONE